# Using Dynamic Neural Networks to Model the Speed-Accuracy Trade-Off in People

**Ajay Subramanian**
New York University
as15003@nyu.edu

**Omkar Kumbhar**
New York University
omkar.kumbhar@nyu.edu

**Elena Sizikova**
New York University
es5223@nyu.edu

**Najib J. Majaj**
New York University
najib.majaj@nyu.edu

**Denis G. Pelli**
New York University
denis.pelli@nyu.edu

## Abstract

Neural networks have been shown to exhibit remarkable object recognition performance. We ask here whether such networks can provide a useful model for how people recognize objects. Human recognition time varies, from 0.1 to 10 s, depending on the stimulus and task. Slowness of recognition is a key feature in some public health issues, such as dyslexia, so it is crucial to create a model of human speed-accuracy trade-offs. This is an essential aspect of any useful computational model of human cognitive behavior. We present a benchmark dataset for human speed-accuracy trade-off in recognizing a CIFAR-10 image [1] from a set of provided class labels. Within a series of trials, a beep sounds at a fixed delay after the target (the desired reaction time), and the response counts only if it occurs near that time. We observe that accuracy grows with reaction time and examine several dynamic neural networks that exhibit a speed-accuracy trade-off as humans do. After limiting the network resources and adding image perturbations (grayscale conversion, noise, blur) to bring the two observers (human and network) into the same accuracy range, humans and networks show very similar dependence on duration or floating point operations (FLOPS). We conclude that dynamic neural networks are a promising model of human reaction time in recognition tasks. Understanding how the brain allocates appropriate resources under time pressure would be a milestone in neuroscience and a first step toward understanding conditions like dyslexia. Our dataset[1] and code[2] are publicly available.

## 1 Introduction

This project benchmarks and models the reaction time of human and neural network object recognition. There have been great advances in understanding and modeling how people recognize objects (see [2, 3]), but less on the timing. An important characteristic of human behavior is the speed-accuracy trade-off, the ability to flexibly trade-off performance for reaction time. An accurate computational model of the human speed-accuracy trade-off would bring us one step closer to better modeling of human physiology and addressing reading deficits such as dyslexia. In Figure 1, we show typical speed-accuracy trade-offs observed in the human data we have collected, when subjects are presented with color, grayscale, noisy, and blurry images. Performance increases when additional time is given

---

[1]See https://osf.io/zkvep/ for dataset.
[2]See https://github.com/ajaysub110/anytime-prediction for code.

Submitted to the 35th Conference on Neural Information Processing Systems (NeurIPS 2021) Track on Datasets and Benchmarks. Do not distribute.

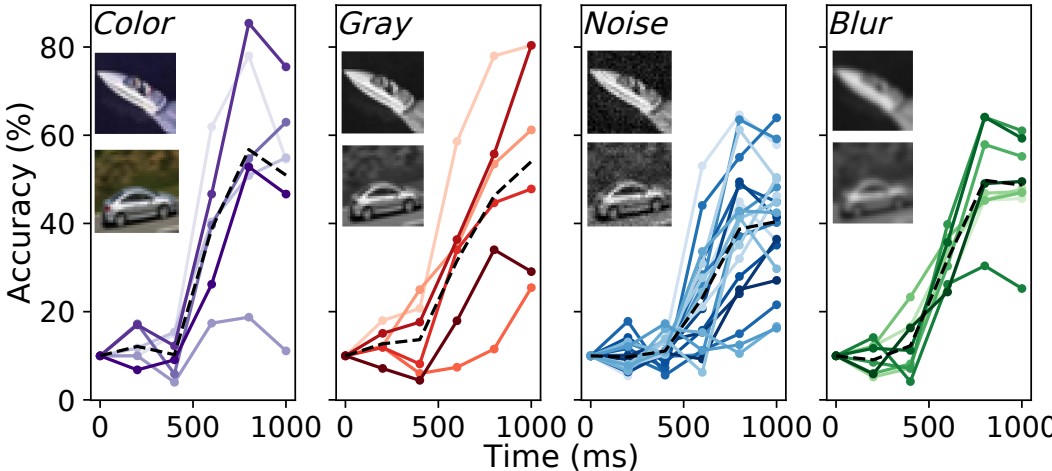

Figure 1: *Sample human performance (accuracy) as a function of allowed reaction time under 4 viewing conditions.* **Left-end:** original colored images (8 observers); **Left-middle:** grayscale-converted images (8 observers) **Right-middle:** Gaussian noise with standard deviation 0.04s (20 observers), **Right-end:** Gaussian blur with standard deviation 1.0s (7 observers). Different colors and opacity of curves represent different observers. Black dotted line is average across observers. In all cases, accuracy tends to increase with response time which indicates the trade-off in accuracy for speed. Accuracy at 0s was not measured and is assumed to be at chance. Best viewed in color.

to the observers. As we will see later, the corresponding curves for a dynamic neural network capture a similar pattern, rising gradually from chance to maximum performance.

As signal strength (e.g., contrast) increases, humans respond more quickly and more accurately, and there is a tight relation between sensitivities measured by accuracy or by reaction time. [4] showed that a diffusion model of perceptual decision making could account for the relation. Humans respond to instructions that change the emphasis on speed vs. accuracy, and can even learn to always respond with a fixed latency [5]. We adopt that paradigm here. For each block of trials the observer is taught to respond at a fixed latency to different perturbation intensities. Each block yields a point in a plot of accuracy vs. latency, and the responses from many blocks trace out the speed-accuracy trade-off. We measure network and human accuracy for the same stimuli and tasks. Reaction time is measured in milliseconds (ms) for the human, and calculated as the number of floating point operations (FLOPs) consumed by the network. The task is to identify the predefined category (1 of 10) of an image from the CIFAR-10 set [1], a collection of natural images commonly used as a computer vision benchmark.

In order to explain the human trade-off, we have analyzed three recent computational models which allow for early exits and adaptive computation as ways to vary computational effort. These strategies are covered in detail in future sections. The first model is a convolutional recurrent neural network (ConvRNN), introduced by [6] which was previously presented as a computational model of speed-accuracy trade-off. This model relies on confidence saturation as an exit strategy to perform dynamic computation. The other two models, MSDNet [7] and SCAN [8], are two popular dynamic depth, anytime prediction models that are used for computer vision and related applications. We present the computation models with degraded stimuli, and measure correlation with timings of the observers to compare their speed-accuracy trade-off patterns. Our results indicate that anytime prediction is a promising model for accuracy and reaction time in human object recognition because it achieves a high correlation with human trade-offs. Our main contributions are:

- We collect and release a benchmark, a speed-accuracy trade-off in human performance, with various image perturbations (grayscale conversion, noise, and blur). This comes from our study of how human observers recognize objects under less than ideal viewing conditions. The speed-accuracy trade-off is an essential property of human object recognition and we encourage further research in designing computational models that can capture it.

- We compare the ability of several networks to capture the speed-accuracy trade-off, and show that an existing dynamic depth neural network (MSDNet [9]) exhibits similar behavior as humans.
- We perform an extensive quantitative comparison between humans and networks, and analyze which models exhibit more human-like performance trade-offs. In doing so we introduce two metrics: performance range, and a correlation metric which ease comparison of model behavior with that of humans.

## 2 Related work

**Measuring the speed-accuracy trade-off in humans.** Given more time people can do better. [5] analyzed the speed-accuracy trade-off in humans on a visual search task, in which observers tried to find a target in an array of distractors. They manipulated task difficulty by adding more and more distractors. Figure 1 shows human object recognition accuracy on CIFAR-10 images as a function of reaction time [1]. [10] proposes a model to predict reaction time in response to natural images. This model is based on statistical properties of natural images and is claimed to accurately predict human reaction time by forming an entropy feature vector. [11] used a drift diffusion model whose drift rate (the rate of accumulation of evidence towards a criterion) was determined by the quality of information to explain lexical decision times and performance (i.e. how rapidly does a person classify stimuli as words or non-words). Reaction time has also been studied in the context of perceptual decision making [4, 12, 13, 14]. [6] is the first to use a neural network as a computational model of the speed-accuracy trade-off, showing that a recurrent neural network (RNN) allows a flexible trade-off between speed and accuracy. Neural networks have also been used to model object recognition [15], temporal dynamics in the brain [16, 17], the ventral stream, i.e., the object recognition neural pathway in human cortex [18], and temporal information [19].

**Dynamic inference.** Dynamic object recognition models adapt their architecture to the challenge of input data to reduce mean cost of inference. There are two classes of dynamic networks. Dynamic width networks, also known as dynamic pruning, use a variable subset of convolution filters to reduce inference cost [7, 20, 21, 22]. Dynamic depth networks perform efficient computation by either early-exiting when their shallow sub-network achieves a high classification confidence [23, 9] or by dynamically skipping layers using residual connections [24, 25]. A more detailed overview of dynamic neural networks and their applications can be found in [26]. In this work, we evaluate the ability of two recent dynamic depth networks [9, 8] to capture the human speed-accuracy trade-off, and compare their performance to existing techniques [6].

## 3 Collecting Behavioral Data

We measured performance and reaction time for human observers performing an object recognition task on images presented with and without perturbation. We assessed the impact of adding color, blur, and noise, The results show a speed-accuracy trade-off (Figure 1) for all three image manipulations. In Sections 4 and 5, we evaluated the ability of dynamic neural networks to model the trade-off between processing speed and accuracy. Our experimental protocol is similar to [5] and is outlined below.

### 3.1 Images

In all experiments, human observers recognized objects in CIFAR-10 images [1], a popular benchmark for neural network analysis, with the default train/test split. This image set contains 50,000 training images and 10,000 test images each of $32 \times 32$ pixels, and has 10 classes: airplane, automobile, bird, cat, deer, dog, frog, horse, ship and truck. Sample images and added perturbations can be seen in Figure 2. We used lab.js [27] and Just Another Tool for Online Studies (JATOS) [28] to present images and collect timed responses from human observers online.

We chose to use CIFAR-10 instead of the popular ImageNet dataset because the 1000 classes in the latter would be too many for our human participants to memorize. Unlike Spoerer et al. [6], we decided against using a subset of ImageNet classes since that would bring in ambiguity of what classes to select. To resolve this issue, Spoerer et al. [6] instead pose a binary classification problem ("animate" vs "inanimate" objects). However, a binary classification task is not representative

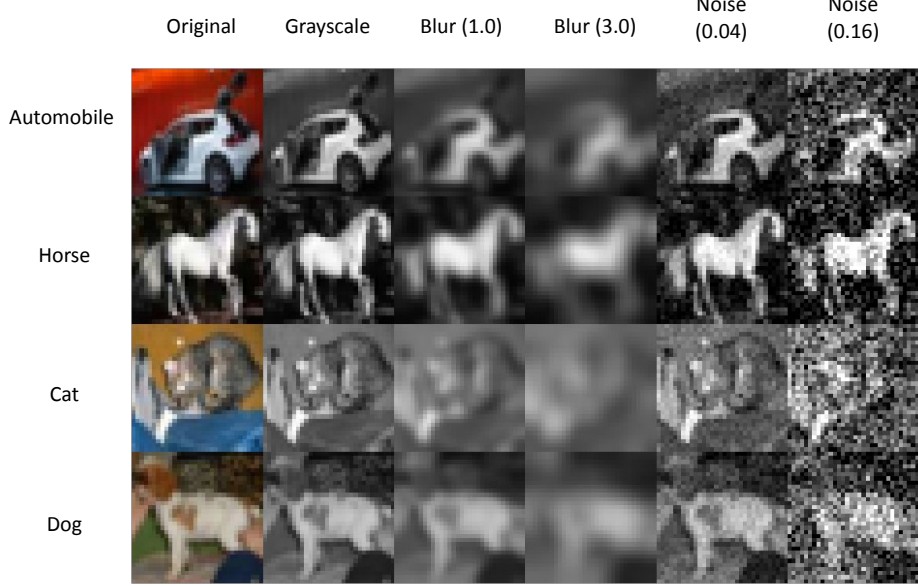

Figure 2: *Example images from the CIFAR-10 dataset [1] along with visualizations of image perturbations considered for human subject experiments – grayscale conversion, image blurring and noise.* By introducing image perturbations, we control the recognition task difficulty. Numbers in parentheses correspond to standard deviations for 0-mean Gaussian distributions.

of general categorization performance because in a binary task an observer may learn to detect the difference between classes rather than actually classify images into one of several classes. Additionally, most ImageNet classes are very specific (eg. "electric ray", "robin", "goldfinch") and hence the method by which a subset of classes is selected would affect human performance.

### 3.2 Observer statistics and data collection

We collected data from 35 observers (23 Male, 12 Female) ranging in age from 24 to 62 years. Each session lasted about an hour. Each observer had a normal or corrected-to-normal vision. The stimuli were presented via JATOS survey via worker links to each observer. Participants were recruited through Amazon MTurk (similar to studies in [29, 2]), and paid $15 for their efforts (to a total of $594 with all fees). A standard IRB approved (IRB-FY2016-404) consent form was signed before collecting the data by each observer, and demographic information was collected.

Table 1: *Summary statistics of collected data on human observers across all experiments.*

| Perturb. | Participants | Avg. Compl. (min) | Questions |
|---|---|---|---|
| Noise | 20 | 57.94 | 1500 |
| Blur | 7 | 53.95 | 1500 |
| Color | 8 | 20.53 | 500 |

Our survey design was based on the previous work by McElree & Carrasco [5], where 4 observers participated in a total of 20 approximately 75 min sessions. At the beginning of each session subjects were instructed that each object category was linked to a particular letter-key: *(A)irplane, a(U)tomobile, (B)ird, (C)at, d(E)er, (D)og, (F)rog, (H)orse, (S)hip* and *(T)ruck*. They were then given a training run of 20 images where they learned the key-class labels getting feedback on the speed of their responses.

Images were interpolated to 190×190 pixels for optimal viewing [30]. A trial consisted of a stimulus image displayed for a fixed amount of time. Since 150 ms is the minimum visual processing time needed to process (recognize) a stimulus [31], the survey was designed on five fixed viewing conditions (blocks) at 200 ms, 400 ms, 600 ms, 800 ms, and 1000 ms with a tolerance of ±100 ms. Outside of these tolerance values, trials were discarded. The survey was designed to control the response time of human observers by asking them to respond in the allotted time distribution. This controls their processing time [5].

To capture variability in observer responses, for noise and blur surveys, each time condition block consisted of 300 trials (1500 trials in total) while the color survey had 100 trials (500 trials in total). At the end of the time-limit for a trial, a beep sounded within 60 ms of which the observer had to enter their category decision via key-press after which feedback was given: if they were quick, slow or perfect while pressing the key.

Observers were asked to place their hands on the keyboard while being aware of the ten identifiers (A: Airplane, C: Cat and so on). Observers were instructed to answer at the beep as fast as possible to fall into the tolerance bounds. Observers were given feedback after every trial and progress was shown in the form of a counter. Pressing the spacebar presented the next stimulus. Before starting the actual survey for data collection, a tutorial of 20 images was displayed to make observers understand the key mapping and get used to the timing protocol. To reduce the length of each experimental session, each observer responded to a randomly selected subset of 1,000 images. This image set was divided into approximately equal chunks across different amounts of perturbation (noise and blur). Figure 1 plots sample human accuracy as a function of reaction time. At 1000 ms, most observers had accuracies about 40% to 50%, except for a few outliers. We created aggregate results across observers to create an average observer, and compared its performance to the computational models.

## 4 Computational models for Speed-Accuracy Trade-Off

In order to test the ability of dynamic neural networks to capture the flexible, adaptive computation that humans exhibit, we analyze three representative models from existing literature. The first two, MSDNet [9] and SCAN [8], both state-of-the-art dynamic depth networks, were originally developed to improve test-time efficiency in computer vision applications. They are promising candidates for our purpose since they are capable of adaptive computation. We compare them against rCNN (which we refer to as ConvRNN) [6], a convolutional recurrent network which was recently developed specifically as a model for human speed-accuracy trade-offs. It should be noted that, due to prior knowledge in humans and other confounding factors, it is difficult to replicate exactly the same training and testing conditions in humans and machines. To partially account for this, we perform trial runs for humans on sample data (see Section 3.2) and test both humans and networks on a variety of perturbation types and strengths. We compare networks with humans, first on the range of performance (accuracy) they can achieve by only varying FLOPs used. Next, we measure their correlation with human behavior under various perturbation conditions to determine if these models can capture the same performance trends that humans exhibit.

### 4.1 Convolutional Recurrent Neural Network (ConvRNN)

ConvRNN [6] exhibits temporal behavior by relying on recurrent connectivity, characteristic of the primate visual system, implemented by adding bottom-up and lateral connections to a feed-forward convolutional network. Lateral connections add cycles inside the feed-forward connectivity allowing for recurrent behavior. This model consists of 7 blocks of recurrent convolutional layers (RCL), followed by a Readout layer to output class predictions. During inference for a given image, the computation used by the model can be dynamically chosen by running the model for a variable number of recurrent cycles. This property allows the network to respond to an input image with a different amount of computation, which we use to represent reaction time.

**Training Details**  Each image was up-sampled to $128{\times}128$ using bi-cubic interpolation to match the input dimensions needed by the network. To prevent overfitting, the model was initialized with pre-trained ImageNet [32] weights and all layers before fully connected layers were frozen for subsequent training. The network was trained to optimize cross-entropy loss over classification targets using Adam optimizer with learning rate 0.005 and epsilon parameter 0.1. L2 regularization was applied throughout training with coefficient of $10^{-6}$. The model was trained for 100 epochs with a batch size of 32.

### 4.2 Multi-Scale Dense Network (MSDNet)

MSDNet [9] implements dynamic inference using multiple early exit classifiers from a feedforward network. Since the exits are all at different depths in the network, classification at each one has a different computational requirement. All exits are placed after blocks of layers and use features from

a common backbone network for classification. A consequence of this is that features deemed useful for each classifier during training interfere with the other classifiers. To resolve this problem, MSDNet proposes two architectural features: multi-scale feature maps, and dense connectivity (realized by using a DenseNet [33] backbone). These properties allow neurons at any layer to access features from any part of the network and at any resolution, thus diminishing the effect of the interference problem. In our experiments, we use a 15-layer backbone network with seven early exit classifiers placed at block intervals of 1-2-4, thus making up a total of seven blocks. Additionally, our setting differs from that used in [9] in terms of the number of scales and bottleneck factor.

**Training details**    During training, MSDNet uses a cumulative cross-entropy classification loss computed over all early exits. The model is trained for 300 epochs and uses a Stochastic Gradient Descent (SGD) optimizer with a learning rate of 0.1 and batch size of 64. Data augmentation based on standard techniques mentioned in [9] are applied: during training, images are horizontally flipped with probability 0.5, normalization based on channel means and standard deviation is also done.

### 4.3   Scalable Neural Network (SCAN)

Similarly MSDNet, SCAN [8] implements dynamic inference using early exit classifiers from a common backbone network. Whereas MSDNet uses multi-scale feature maps and dense connectivity to solve the issue of interference between early and late classifiers, SCAN uses an encoder-decoder attention mechanism in each exit network. This allows each exit to "focus" only on features relevant for classification at a specific depth of the backbone. The attention network produces a binary mask which is added to the backbone feature map, after which a Softmax layer predicts a class label. In our experiments, we use three variants of SCAN, each with a different backbone architecture: SCAN-R18 with ResNet-18, SCAN-R34 with ResNet-34 and SCAN-R9 with ResNet-9. Each of these uses four early exits and a final ensemble output which uses all early exit features for prediction. Thus, for a given input, the network outputs five class predictions, each requiring different amount of computation time/effort.

**Training Details**    During training, a loss function that combines a cross-entropy term (for classification) and a self-distillation term is computed and summed over all exits. The self-distillation helps improve performance by encouraging a low KL-divergence between the exit outputs and final output distributions, and is controlled using a self-distillation coefficient. In our experiments, SGD with a fixed learning rate of 0.1 and momentum factor of 0.9 is used to optimize network parameters. All variants of SCAN are trained using the same optimizer settings, with a self-distillation coefficient of 0.5 and with a batch size of 128, for 200 epochs.

### 4.4   Contrast adjustment

We found that, in general, the networks were more accurate than human observers when presented with images (see Section 5.1) of the same noise levels. To match human performance, we applied additional noise to images input to networks, which resulted in the application of noise levels that fell outside of the distribution of the image pixels. Therefore, a clipping value, which changes the pixel distribution, was required to display the noisy images. Since our primary goal is to mimic human performance with the networks, we adjusted the contrast of the images observed by networks by 10% (see Figure 2 for examples). Contrast adjustment removed the need for image clipping and brought the result ranges from noise from neural networks closer to

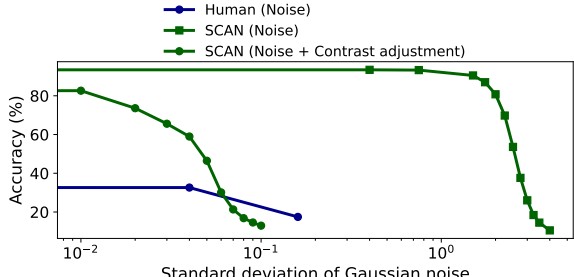

Figure 3: *Effect of contrast adjustment for the case of SCAN.* Contrast adjustment allows us to **a.** bring network performance to the same range as human performance, and **b.** increase task difficulty for network, to avoid clipping in case of high noise levels.

those produced by human observers. We compared the performance of MSDNet [8], the top perform-

ing network, on original and contrast-adjusted images with and without noise, and found that the latter produced more human-like responses in networks than the former.

# 5 Results and discussion

We now study how well human response patterns are matched with results from our computational models. Specifically, we analyze the performance ranges exhibited by each model type and correlate model performance with human response slopes.

## 5.1 Comparing human and model performance range

We analyze and compare human and neural network performance on grayscale CIFAR-10 images in Figure 4 which shows the range of accuracies shown by each model and the human average.

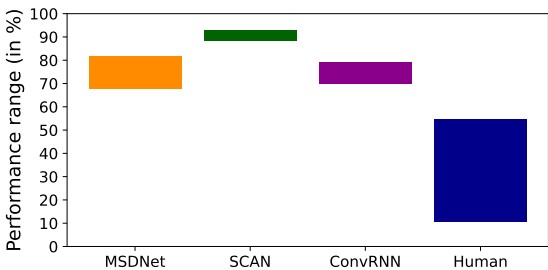

Figure 4: *Performance range of neural networks trained and evaluated on grayscale CIFAR-10 images, and comparison with human average*. The neural networks exhibit higher accuracies but significantly smaller performance range than human observers.

We find that the accuracies achieved by all networks greatly exceed that of human observers (by $> 15\%$). On the other hand, the *performance range* (i.e., difference between maximum and minimum accuracies) is much higher in humans ($44.22\%$) than in networks. Across the neural network models, MSDNet [9] achieved the highest performance range ($13.87\%$), followed by ConvRNN [6] ($9.02\%$), and finally, SCAN [8] ($4.34\%$). The large difference in performance range between humans and networks is primarily because networks achieve high classification accuracies even with low computational effort i.e. the task is trivial. Larger performance ranges can therefore be obtained by reframing the task to make it more challenging.

## 5.2 Varying task difficulty using image perturbations

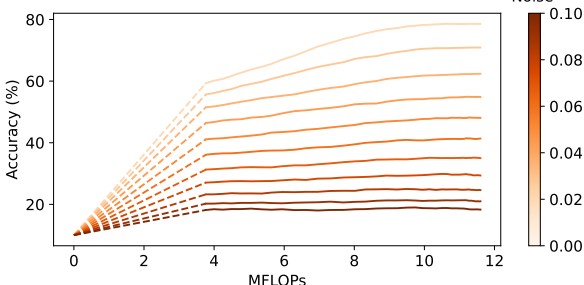

Figure 5: *MSDNet accuracy vs MFLOPs for various values of test image noise.* Each curve corresponds to different Gaussian noise standard deviation $\in [0, 0.1]$, as shown by the colorbar. Performance at 0 MFLOPs is taken to be at chance ($10\%$), attained by any fixed response, and dotted lines extrapolate measured data points to this value. The dotted lines bridge the catastrophic failure of MSDNet, which cannot provide any useful answer at all with less than about 3.5 MFlops. The model was trained with fixed random batch noise $\in [0, 0.05]$

Humans adapt effortlessly to a wide range of task demands. Here, we explore how well machines can do this by comparing the performance range of both humans and networks on perturbed images. Noise in perception experiments is used for assessing unpredictable variation in some aspect of stimulus [34], and we attempt to model the same effect in our experiments. We modify the recognition task by adding noise and blur to make it more challenging, and then analyze the effect. Image perturbations are useful for bench-marking human performance [35, 36]. Additionally, CIFAR-10 is a relatively simple dataset for deep networks and risks getting a ceiling effect. Adding noise and blur to images makes the task more difficult, thus resolving this issue. Figure 5 shows MSDNet's trade-off curves under various amounts of test-time image noise. It can be seen that at zero noise, lowering computation below the lowest possible number of FLOPs would result in a catastrophic drop from 60% to chance. This is unlike humans whose performance drops more gracefully as allowed reaction time is lowered (Figure 1).

(a) Evaluation with noise. Human performance is considered for three noise patterns applied to images, distributed as Gaussian noise with zero mean and standard deviation in {0, 0.04, 0.16}.

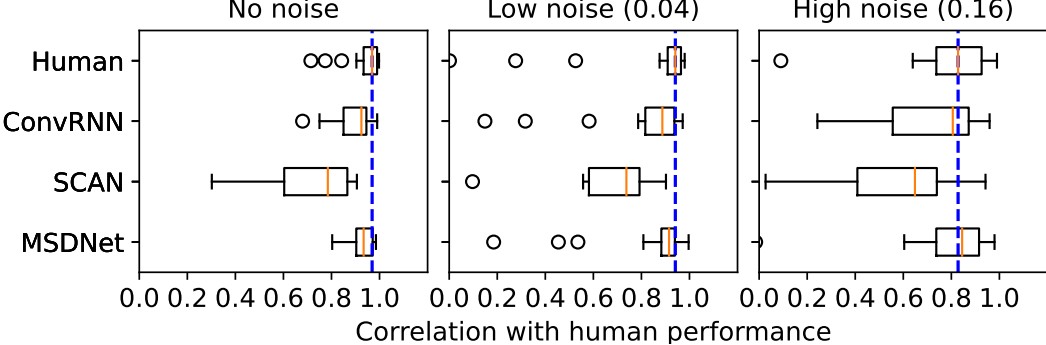

(b) Evaluation with blur. Human performance is considered for three blur patterns applied to images, distributed as Gaussian blur with zero mean and standard deviation in {0, 1.0, 3.0}.

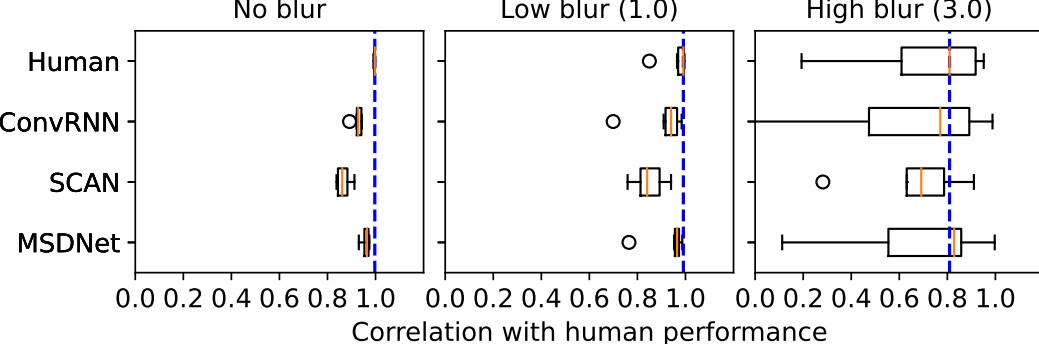

Figure 6: *Correlations of networks with human performance, evaluated across different levels of noise & blur perturbation.* For fair comparison, the level of perturbations used during training is same across all networks. During inference for each network, noise/blur level that elicits highest correlation with humans is found and shown above. MSDNet achieves the highest correlation with human observers in all testing scenarios. Orange bars represent median correlation value. Vertical blue line is an extension of the median correlation of humans with each other. Standard deviation for all correlations is shown.

We correlate network performance to average human performance at varying levels of noise or blur, and report Pearson's *r* correlation coefficients in Figure 6a. To obtain an upper bound on correlations, we also correlate each human observer to the average human observer. Unlike previous work [6] which correlates reaction time for prediction, we report the correlation of *performance as a function of reaction time*. This metric captures both performance and reaction time and hence allow for a more robust evaluation of the speed-accuracy trade-off exhibited by humans and models.

For blur, we find that the MSDNet [9] achieves the highest correlation to humans, followed by ConvRNN [6] and SCAN [8] while for noise, correlations of MSDNet and ConvRNN are both similar and higher than SCAN. When comparing SCAN [8] models with different backbones, we find that decreasing the ResNet [37] backbone to ResNet-9 decreases the correlation. Similarly, choosing an over-parametrized ResNet-34 also adversely affects correlation. It is important to point out the need for much higher noise to bring the network accuracy down to human performance. This indicates that the neural networks are more tolerant to noise than human observers once trained with noisy images.

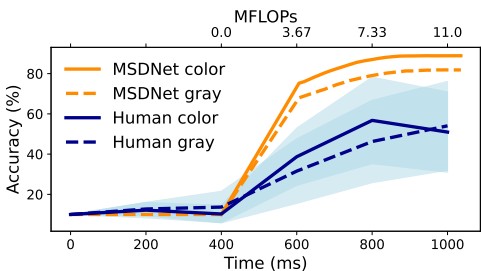

Figure 7: *Evaluation of the effect of color on network and human performance. Color does not significantly affect the recognition performance of either humans or MSDNet [9] model.* Accuracy at 0 Time/MFLOPs was not measured and assumed to be at chance. Linear scaling was used to bring MFLOPs (F) into the same range as Time (T) $[F = 11/600(T - 400)]$. Blue bounding areas represent standard deviation of performance across humans.

## 5.3 Influence of color

We evaluated the effect of color on human and network performance, and reported results in Figure 7. We found that color improves the recognition accuracy for both humans and neural networks by about 5% in both cases. However, the performance range and patterns of improvement when given additional processing resources stayed the same. We conclude that the addition of color did not influence the results reported in other experiments in the paper.

## 5.4 Evaluating the effect of the number of network parameters

The success of modern neural networks is tied to their large number of parameters, a natural question to ask is whether the number of parameters has an effect on human-network correlation. In Figure 8, we evaluate the effect of having differently sized backbone networks on the MSDNet model performance. MSDNet uses a custom architecture based on DenseNet, therefore, we have modified parameters in the configuration without significant updates to the architecture. In general, we found that changing network size did not improve correlation with human performance.

For SCAN [8] and MSDNet [9], we found that networks with deeper backbones (SCAN-R34 and MSDNet-L) exhibit a higher correlation (more human-like speed-accuracy trade-off) compared to human observers. However, this improvement is not significant and does not indicate a monotonic relationship between correlation and backbone parameter count.

Paired 2-tailed t-tests showed that all correlation comparison results mentioned above are significant (p $\leq 0.05$, corrected). Bonferroni correction was used to correct for multiple comparisons. Additionally, our human behavior dataset was deemed to be large enough to draw all previously made inferences, using a sample-size determination test (with p $\leq 0.05$, power = 80%).

## 6 Conclusion

Speed-accuracy trade-off is an essential feature of human performance that is difficult to explain with current computational models of object recognition. We present a benchmark for timed object recognition. Observers were asked to recognize objects in degraded images with a time constraint, and showed a speed-accuracy trade-off. We also show that dynamic depth neural networks are a realistic model of the speed-accuracy trade-off in object recognition. To compare various networks

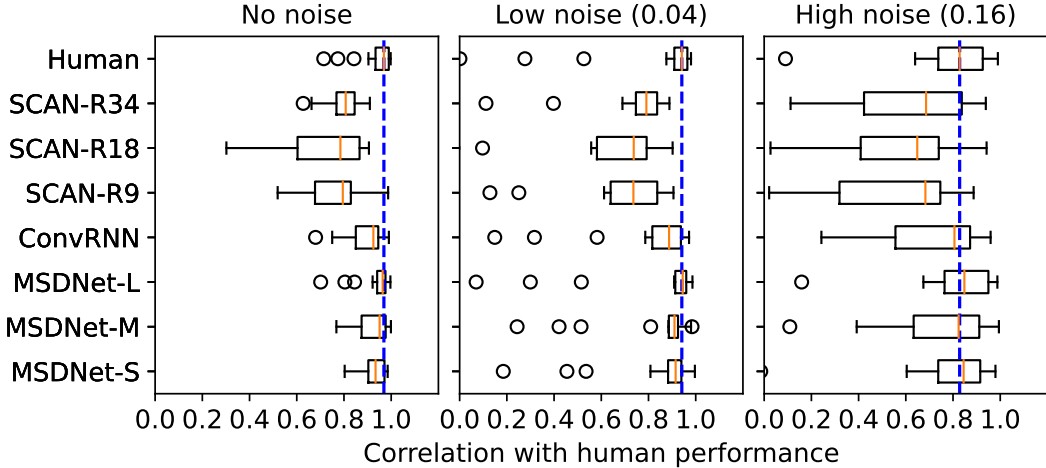

Figure 8: *Evaluation of the effect of the network backbone architecture to correlation with human performance*. SCAN-R18 and MSDNet-S are the versions used in earlier experiments. Backbone notation: SCAN (R34 - ResNet-34, R18 - ResNet-18, R9 - ResNet-9), MSDNet (L - Large, M - Medium, S - Small).

with humans, we propose two metrics: performance range, and correlation of performance as a function of reaction time, which together capture both the magnitude of trade-off as well as similarity with human trade-offs. One of the considered models, MSDNet [9], gives a better account than previous attempts [6], without the need for recurrence. When faced with noise or blur, machine performance deteriorates in a quantitatively similar fashion as human performance. When trained with noise, it shows a maximum of $94\%$ correlation with human performance and $96\%$ when trained with blur. Finally, we test the effect of network backbone architecture and determine that correlation to human performance does not necessarily increase with additional parameters.

While dynamic networks succeed in showing some speed-accuracy trade-off, their range is less than what humans achieve. The average human performance range is $44.22\%$ while the best network, MSDNet trained with noise achieves only $19.24\%$. With high perturbation strength, humans stumble and machines fall. This motivates future work that aims to build neural networks that can better match the flexibility and adaptability of human object recognition. Work in this direction is important for achieving a better understanding of human decision making and for deployment of machine learning systems in time-sensitive applications.

Slow (dyslexic) and fast readers show a marked difference in speed-accuracy trade-off during reading. Slow readers need more time on average to achieve the same reading performance as a fast reader. Several behavioral results demonstrate this difference in speed-accuracy trade-off, but no satisfactory computational models have been developed. The dataset and benchmark proposed in our paper are an attempt to encourage work seeking to develop models that demonstrate a more human-like speed accuracy trade-off.

The primary focus of current machine learning research has been on improving peak performance. When the allowed computational effort ($\propto$ reaction time) is restricted, human performance drops gracefully while neural network performance fails catastrophically. In other words, the problem with machines performing consistently well over all FLOPs values is that lowering their computational resources below a certain point will cause a huge drop in performance resulting in near-chance performance. In time-sensitive applications like autonomous navigation, catastrophic failure due to time limitation is unacceptable. Humans have this ability and efforts to introduce it to machine learning models are just beginning.

The applications of the above-described technology have potential benefits (addressing public health concerns and biases in computational models) and risks (from malicious data augmentation to surveillance). We believe that these concerns are shared in general by machine learning applications, and are outside the scope of this work.

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
