# OpenReview forum: "Using Dynamic Neural Networks to Model the Speed-Accuracy Trade-Off in People"
_NeurIPS.cc/2021/Track/Datasets_and_Benchmarks/Round1 — Submitted to NeurIPS 2021 Datasets and Benchmarks Track (Round 1)_

### Official Review · Reviewer_Qqdq · 2021-06-30
**Neat idea, weak/unclear contribution**

**Rating:** 3
**Confidence:** 2

**Strengths:**

1. **Robust, well-documented data collection.** The data collection procedure appears to be very carefully designed based on prior work and has no obvious flaws in internal validity. (I am not sure about the standards for external validity in human cognition studies re: sampling bias - the sample seems to be primarily male.)
2. **Clever, interdisciplinary approach.** Controlling the recognition task difficulty with image perturbations was a good idea! Prima facie this seems to be a good solution for mimicking the human trade-off, and I like the idea to use neural networks to model this problem.

**Weaknesses:**

1. **[Major] Potentially incomplete analysis.** I am concerned that Pearson's $r$ might not be the right choice for measuring correlations, especially since the results are not reported with a clearly explained standard error and the interpretation is not clear. Do you obtain any confidence bounds? What distributional assumptions are made on the data? For example, Figure 5 shows boxplots of the correlation coefficients, but doesn't provide any standard errors as far as I can tell. I would encourage the authors to clarify the statistical significance of their results, noting that Pearson's r is not robust and could be misleading if there are outliers.

EDIT (07/16/2021): The authors added significance tests and std bounds (but still no confidence intervals) - but the paper still doesn't mention distributional assumptions and significance results are not reported with each experiment (as far as I can tell the tests were performed only for Figure 8).

2. **[Major] Motivation and relevance.** The authors state several times that the speed accuracy trade-off is "an essential property of human object recognition." Without a background in this area, it's difficult to tell exactly why it's important, except for a vague alluson to dyslexia and other deficits.

    If the main contribution is a benchmark to open up research in this area, it's important to clearly explain why this research and these results matter to biology/physiognomy/psychology. I only see one minor contribution (L281), which claims that MDSNet better models the speed accuracy trade-off than a previous attempt, but the limitations in the statistical analysis undermine this claim.

    It's especially important to demonstrate a biological contribution since there isn't a clear contribution to machine learning. For example, in L290-291, the authors state that future work in object classification should try to match the flexibility of human object recognition based on the greater performance range of human participants. But it's not clear why performance range is desirable - if machines are performing consistently well, isn't that a desirable outcome? Is there some evidence that performance range corresponds to generalizability or robustness? There is already lots of research on robustness to data augmentation in computer vision.

EDIT (07/16/2021): Conversations with the authors have helped clarify this point. I am still concerned that the authors are split between two contributions: explaining human cognition with computational models (probably more appropriate for another track/venue), and benchmarking NNs for graceful failure under computational constraints. The original submission has very little focus on the ML contribution. The authors make references to proposed changes emphasizing the latter contribution in the Intro, Related Work, and Results, but I don't see those in the manuscript yet - only a paragraph in the discussion without the new references mentioned in our discussions.

**Additional Feedback:**

I think this is an interesting avenue of research, but I'm having a difficult time understanding why the results are relevant to machine learning or the study of human cognition. Where can we go from here?

EDIT (07/16/2021): The authors have clarified both points. I am unqualified to judge the contribution to CogSci but the authors' comments are compelling. I am less convinced about the contribution to ML research, given that the manuscripts I read focus mostly (in their motivations, methods, and results) on modeling/explaining human behavior, not benchmarking ML models.

**Clarity:**

The paper is mostly clear, but in the introduction it is difficult to follow and relies heavily on jargon. For example, in lines 42-44, the authors make reference to early exits and adaptive composition. Why are these important for explaining human speed-accuracy trade-offs? What do these terms mean? (The explanation doesn't come until later.)

The description of methods is especially clear - well done.

The figure captions are fairly sparse - for example, Figure 6 shows a blue bounding area but doesn't describe what it represents (a standard error?).

EDIT (07/16/2021): These concerns have been resolved by additional clarification and more detailed captions.

**Correctness:**

The methods and models are thoroughly described and appear to be correctly chosen and parameterized. Without further knowledge of similar cognition studies it is difficult to find flaws in the experimental design other than the ones detailed in the **Weaknesses** section.

**Documentation:**

I don't see any clear gaps in the documentation. I was able to access the dataset URL easily and the site is well documented - great work. The paper and website together provide instructions for reproducing the study.

**Ethics:**

This study appears to have IRB approval and compensates participants. I don't see any obvious ethical concerns, so long as this is true. As far as I can imagine there are no concerning uses for this benchmark.

**Relation To Prior Work:**

The authors describe several relevant studies and position their work as an application of the field of dynamic automated object recogition to the problem of modeling human object recognition.

**Summary And Contributions:**

This paper proposes a object classification benchmark dataset for modeling the speed-accuracy trade-off in human cognition. The authors use the benchmark to compare human cognition to neural network behavior, providing both a model for the human speed-accuracy trade-off and recommendations for computer vision research.

By way of framing this review: I am familiar with computer vision research in general, but I have very little knowledge of human cognition research of this kind.

---

> ### Author Response · Authors · 2021-07-12
> **Response Part 1**
>
> __*Robust, well-documented data collection. The data collection procedure appears to be very carefully designed based on prior work and has no obvious flaws in internal validity. (I am not sure about the standards for external validity in human cognition studies re: sampling bias - the sample seems to be primarily male.)*__
>
> __*Clever, interdisciplinary approach. Controlling the recognition task difficulty with image perturbations was a good idea! Prima facie this seems to be a good solution for mimicking the human trade-off, and I like the idea to use neural networks to model this problem.*__
>
> *Thank you.*
>
>
> __*[Major] Potentially incomplete analysis. I am concerned that Pearson's might not be the right choice for measuring correlations, especially since the results are not reported with a clearly explained standard error and the interpretation is not clear. Do you obtain any confidence bounds? What distributional assumptions are made on the data? For example, Figure 5 shows boxplots of the correlation coefficients, but doesn't provide any standard errors as far as I can tell. I would encourage the authors to clarify the statistical significance of their results, noting that Pearson's r is not robust and could be misleading if there are outliers.*__
>
> *Done.* As requested, we added a test for statistical significance. We ran paired, 2-tailed t-tests using p = 0.05 (with Bonferroni correction, similar to [6]) to evaluate whether the difference in correlation between different models is significant and text in the paper has been updated accordingly. Our main conclusion from the correlation plots still hold: MSDNet, an anytime prediction network shows human-like behavior similar to rCNN (ConvRNN) from Spoerer et al. [6] which was developed specifically as a model for the human speed-accuracy tradeoff. This is now mentioned in lines 296-299 of the updated paper.
>
> *Clarification.* Our purpose was to compare other networks with the most human-like in [6] over a wider range of performance. Of course, we compared using the same statistic, namely Pearson correlation between model and average human performance. That is necessary to our goal. We agree that Pearson correlation may not tell the whole story, and we are interested in exploring other summary statistics in future work.

---

> > ### Author Response · Authors · 2021-07-12
> > **Response Part 2**
> >
> > __*[Major] Motivation and relevance. The authors state several times that the speed accuracy trade-off is "an essential property of human object recognition." Without a background in this area, it's difficult to tell exactly why it's important, except for a vague alluson to dyslexia and other deficits.*__
> >
> > __*If the main contribution is a benchmark to open up research in this area, it's important to clearly explain why this research and these results matter to biology/physiognomy/psychology. I only see one minor contribution (L281), which claims that MDSNet better models the speed accuracy trade-off than a previous attempt, but the limitations in the statistical analysis undermine this claim.*__
> >
> > __*It's especially important to demonstrate a biological contribution since there isn't a clear contribution to machine learning. For example, in L290-291, the authors state that future work in object classification should try to match the flexibility of human object recognition based on the greater performance range of human participants. But it's not clear why performance range is desirable - if machines are performing consistently well, isn't that a desirable outcome? Is there some evidence that performance range corresponds to generalizability or robustness? There is already lots of research on robustness to data augmentation in computer vision.*__
> >
> > *Done.* These are good points which we address here and in the manuscript (Section 6, lines 318-334). Modeling the speed-accuracy tradeoff is important for deployment of machine learning systems in time-sensitive applications and to achieve a better understanding of human decision making.
> >
> > Slow (dyslexic) and fast readers show a marked difference in speed-accuracy tradeoff during reading. Slow readers need more time on average to achieve the same reading performance as a fast reader (Marinelli et al. 2016, PLOS ONE). Several behavioral results demonstrate this difference in speed-accuracy tradeoff, but no satisfactory computational models have been developed. The dataset and benchmark proposed in our paper are an attempt to encourage work seeking to develop models that demonstrate a more human-like speed accuracy tradeoff.
> >
> > The focus of current machine learning research has been on improving peak performance. When the allowed computational effort (~ reaction time) is restricted, human performance drops gracefully while neural network performance fails catastrophically. In other words, the problem with machines (MSDNet, rCNN, SCAN for example) performing consistently well over all FLOPs values is that lowering their computational resources below a certain point will cause a huge drop in performance from near say, 70-80% to chance. In time-sensitive applications like autonomous navigation, catastrophic failure due to time limitation is unacceptable. Humans have this ability and efforts to introduce it to machine learning models are just beginning.
> >
> > We added this background to the Introduction.
> >
> >
> > __*The paper is mostly clear, but in the introduction it is difficult to follow and relies heavily on jargon. For example, in lines 42-44, the authors make reference to early exits and adaptive composition. Why are these important for explaining human speed-accuracy trade-offs? What do these terms mean? (The explanation doesn't come until later.)*__
> >
> > *Done.* Good point. Now spelled out in the updated version of the Introduction.
> >
> > __*The figure captions are fairly sparse - for example, Figure 6 shows a blue bounding area but doesn't describe what it represents (a standard error?).*__
> >
> > *Done.* Captions for Figure 5 and 6 have now been updated with a better explanation.

---

### Official Review · Reviewer_B3mW · 2021-07-03
**Intriguing observations about human speed-accuracy tradeoff**

**Rating:** 6
**Confidence:** 5
**Clarity:** The paper is well written.

**Strengths:**

-- It is very useful and important for the field to compare neural networks with human performance
-- Especially relevant toward understanding visual computations is the processing time of visual information, which the authors aim to study here


**Weaknesses:**

-- It is hard to compare studies with different datasets, different chance levels, etc. Therefore, the following comparisons should be interpreted with caution. Here are a couple of examples:
Majaj et al J. Neuroscience 2015 (first author is one of the authors in the current paper). The authors report d’ and 8-way categorization instead of classification accuracy with 10 classes (and completely different images), but the authors write: “The image presentation time was chosen based on results showing that core object recognition performance improves quickly over time such that accuracy for a 100 ms presentation time is within 92% of performance at 2 s (see Fig. S2 in Cadieu et al., 2014). Results were very similar, with slightly shorter (50 ms) or longer (200 ms) viewing duration.” Similar observations from the same lab in monkeys are reported in Rajalingham et al J Neuroscience 2018. These conclusions are completely different from the ones in the current study.
Tang et al PNAS 2018. This is 5-way categorization. Stimuli presented for 25-150 ms. Subjects are almost at 100%, even with 25 ms presentation time. Even with heavy occlusion, the subjects do much better than in the current study.
Kirchner et al Vision Research 2006. Binary classification of animal or not. The subject with the worst accuracy had 77.1% (best accuracy was 98.7%) with 120 ms presentation time.
Geirhos et al arXiv 2018 (see also recent Geirhos et al arXiv 2021). 16 classes, 200 ms presentation time. Accuracy for both color and grayscale images was about 90%. Even after addition of large amounts of noise (similar types of noise but much more noise than in the current study), performance was always much better than in the current study.
There are many more studies, but I include just a few above to keep the discussion concrete. My understanding from Figure 1 is that chance is 10%. Thus, subjects seem to be at chance up to approximately 400 ms. Some subjects are even at chance at 1000 ms!!! This seems to be completely incompatible with the extensive literature showing basically ceiling performance with 100-200 ms stimulus presentation time.
-- The authors are experts in the field and the last authors is one of the key creators of the famous psychophysics toolbox, arguably one of the best ways to control timing in stimulus presentation. We are all indebted to his contributions. Thus, I am quite convinced that there is something about the current study that I am failing to grasp. But I cannot figure out what it is.
-- Perhaps the CIFAR-10 images are heavily pixelated and therefore it is extremely hard to recognize them and this is even worse than the noise levels in Geirhos et al 2018, 2021 or the occlusions in Tang et al 2018?
The authors do not report the size in visual angle of the stimuli (perhaps because it is very hard to control in online experiments). Perhaps subjects were doing the tests on their phone and the stimuli were incredibly small?
How do the authors control stimulus timing in online experiments? Perhaps the x-axis does not accurately reflect the actual stimulus presentation time? But it would have to be off by almost an order of magnitude to make the results consistent with the literature.
It would be nice to show actual psychophysics results in the lab to compare with the online measurements. Perhaps the online subjects are watching Netflix, and Snapchat while performing the tasks?


**Additional Feedback:**

I gave this study a very low ranking but I am willing to radically change my ranking if the authors can explain why their results are so different from everything else published in the field.



**Correctness:**

As alluded to above, the results make no sense to me. I apologize to the authors for not being able to figure out what is going on. I hope that their rebuttal will clarify

**Documentation:**

The documentation is clear.

**Ethics:**

No ethical concerns

**Relation To Prior Work:**

My main concern is the relationship between the results and previous work as outlined above. I fail to understand how the current results can be consistent with an extensive body of literature documenting human (and monkey) visual recognition behavior.

**Summary And Contributions:**

This study focuses on studying human recognition performance of CIFAR-10 images, including modified versions of those images, as a function of the stimulus presentation time. The study further compares human performance against recognition by different types of neural networks. I am quite perplexed by the results, which do not match my intuitions and expectations based on previous studies. Extensive studies show that humans can recognize objects quite well in about 100-200 ms presentation time (see references below) but here the authors report chance performance up to 400 ms and even rather poor performance at 1000 ms presentation times. Given that the authors are top-notch scholars in the field, there must be something quite basic and deep that I am misunderstanding about the paper but I am afraid that I fail to see what it is.

---

> ### Author Response · Authors · 2021-07-12
> **Response Part 1**
>
> __*It is hard to compare studies with different datasets, different chance levels, etc. Therefore, the following comparisons should be interpreted with caution. Here are a couple of examples: Majaj et al J. Neuroscience 2015 (first author is one of the authors in the current paper). The authors report d’ and 8-way categorization instead of classification accuracy with 10 classes (and completely different images), but the authors write: “The image presentation time was chosen based on results showing that core object recognition performance improves quickly over time such that accuracy for a 100 ms presentation time is within 92% of performance at 2 s (see Fig. S2 in Cadieu et al., 2014). Results were very similar, with slightly shorter (50 ms) or longer (200 ms) viewing duration.” Similar observations from the same lab in monkeys are reported in Rajalingham et al J Neuroscience 2018. These conclusions are completely different from the ones in the current study. Tang et al PNAS 2018. This is 5-way categorization. Stimuli presented for 25-150 ms. Subjects are almost at 100%, even with 25 ms presentation time. Even with heavy occlusion, the subjects do much better than in the current study. Kirchner et al Vision Research 2006. Binary classification of animal or not. The subject with the worst accuracy had 77.1% (best accuracy was 98.7%) with 120 ms presentation time. Geirhos et al arXiv 2018 (see also recent Geirhos et al arXiv 2021). 16 classes, 200 ms presentation time. Accuracy for both color and grayscale images was about 90%. Even after addition of large amounts of noise (similar types of noise but much more noise than in the current study), performance was always much better than in the current study. There are many more studies, but I include just a few above to keep the discussion concrete. My understanding from Figure 1 is that chance is 10%. Thus, subjects seem to be at chance up to approximately 400 ms. Some subjects are even at chance at 1000 ms!!! This seems to be completely incompatible with the extensive literature showing basically ceiling performance with 100-200 ms stimulus presentation time. -- The authors are experts in the field and the last authors is one of the key creators of the famous psychophysics toolbox, arguably one of the best ways to control timing in stimulus presentation. We are all indebted to his contributions. Thus, I am quite convinced that there is something about the current study that I am failing to grasp. But I cannot figure out what it is. -- Perhaps the CIFAR-10 images are heavily pixelated and therefore it is extremely hard to recognize them and this is even worse than the noise levels in Geirhos et al 2018, 2021 or the occlusions in Tang et al 2018?*__
>
> *Done.* We explain here why performance values reported in our paper are lower than those from other papers. There are several differences between our experimental procedures and those in the papers you referenced:
>
> * *The Carrasco & McElree speed-accuracy-tradeoff (SAT)* paradigm was a major advance in tracking the improvement of performance with time. Allowing observers to respond when they feel like and then sorting into bins produces confounds that make the data hard to analyze because observers tend to take longer on harder trials. Instead they trained observers to respond at a fixed time (different in each block), so measured performance is not confounded with trial-by-trial difficulty. Our use of their paradigm makes our results much easier to analyze.
>
> * *Stimulus duration:* In most studies of the effect of timing in object recognition (including those cited in the review above), each trial’s stimulus presentation and choice selection are separate steps. They report various stimulus durations (e.g. 100-2000 ms in [1], 100 ms in [2], 25-150 ms in [3], 200 ms in [4]), after which the observers were allowed to take as much time as needed to make their selection (>=300 ms in [1], <=1000 ms in [2], >=500 ms in [3], <=1500 ms in [4]). In our experiments (the SAT paradigm), each trial is one step. The image stays on until the observer responds by pressing a key. Thus our reported reaction times include all the time between stimulus onset and key press. Our observers had very little time to respond, compared to typical object recognition studies, and, in fact, we made sure to restrict the time enough to reduce performance to near chance.
>
> * *Image resolution:* In our experiments, we use 32x32 resolution CIFAR-10 images which are highly pixelated and hence hard to classify for human subjects. In contrast, the papers referenced in your comments use much higher resolution images (256x256 in [1], [2], [3] and 128x128 in [4]) and hence are much easier to classify.
>
> * *Other minor differences:* Some of the previous studies [2] are N categories but each trial involves only a binary decision. Ours require the observer to make a ten-way categorization in each trial. Also, [3] uses only 5 classes while our tasks use 10.

---

> > ### Author Response · Authors · 2021-07-12
> > **Response Part 2**
> >
> > __*The authors do not report the size in visual angle of the stimuli (perhaps because it is very hard to control in online experiments).*__
> >
> > *Done.* All experiments were conducted online through MTurk. We did not measure viewing distance or screen size (cm) online, but we have estimated it to be to be roughly 57 cm. Similarly we estimate the size in cm of the 190x190 pixel image to be 4x4 cm, subtending 4x4 deg. We have added this estimate to Methods.
> >
> > __*Perhaps subjects were doing the tests on their phone and the stimuli were incredibly small?*__
> >
> > *Explanation.* It would be very difficult for subjects to attempt the test on their phone since they are required to make choices using a keyboard. Even if they managed to find a way to do so, it wouldn’t affect performance since observers hold phone much closer.
> >
> > __*How do the authors control stimulus timing in online experiments? Perhaps the x-axis does not accurately reflect the actual stimulus presentation time? But it would have to be off by almost an order of magnitude to make the results consistent with the literature.*__
> >
> > *Done.* We used lab.js software. It reliably gives accurate timing in benchmark evaluation of online testing packages (Bridges et al. 2020 [7]), better than 5 ms trial-to-trial variation in stimulus duration and better than 10 ms trial-to-trial variation in reaction time, across many operating systems and browsers. All of the online packages with accurate timing achieve it by running javascript locally on the browser of the participant’s computer, so there is no use of the internet within each trial. We have added this detail to the Methods.
> >
> > __*It would be nice to show actual psychophysics results in the lab to compare with the online measurements. Perhaps the online subjects are watching Netflix, and Snapchat while performing the tasks?*__
> >
> > *Explanation.* Ha! The SAT task is VERY demanding. Each response must occur in a narrow time window, or is rejected. This requires practice to attain, and intense concentration to sustain. There is no possibility that our participants were dual tasking. We have previously collected very similar data from experiments with in-person testing [5] and observed no significant differences with data collected online in our present study. We have added a mention of this to the Methods.

---

> > > ### Author Response · Authors · 2021-07-12
> > > **References**
> > >
> > > [1] Majaj, Najib J., Ha Hong, Ethan A. Solomon, and James J. DiCarlo. "Simple learned weighted sums of inferior temporal neuronal firing rates accurately predict human core object recognition performance." Journal of Neuroscience 35, no. 39 (2015): 13402-13418.
> > >
> > > [2] Rajalingham, Rishi, Elias B. Issa, Pouya Bashivan, Kohitij Kar, Kailyn Schmidt, and James J. DiCarlo. "Large-scale, high-resolution comparison of the core visual object recognition behavior of humans, monkeys, and state-of-the-art deep artificial neural networks." Journal of Neuroscience 38, no. 33 (2018): 7255-7269.
> > >
> > > [3] Tang, Hanlin, Martin Schrimpf, William Lotter, Charlotte Moerman, Ana Paredes, Josue Ortega Caro, Walter Hardesty, David Cox, and Gabriel Kreiman. "Recurrent computations for visual pattern completion." Proceedings of the National Academy of Sciences 115, no. 35 (2018): 8835-8840.
> > >
> > > [4] Geirhos, Robert, Carlos R. Medina Temme, Jonas Rauber, Heiko H. Schütt, Matthias Bethge, and Felix A. Wichmann. "Generalisation in humans and deep neural networks." arXiv preprint arXiv:1808.08750 (2018).
> > >
> > > [5] Kumbhar, Omkar, Elena Sizikova, Najib Majaj, and Denis G. Pelli. "Anytime prediction as a model of human reaction time." arXiv preprint arXiv:2011.12859 (2020).
> > >
> > > [6] Spoerer, Courtney J., Tim C. Kietzmann, Johannes Mehrer, Ian Charest, and Nikolaus Kriegeskorte. "Recurrent neural networks can explain flexible trading of speed and accuracy in biological vision." PLoS computational biology 16, no. 10 (2020): e1008215.
> > >
> > > [7] Bridges, D., Pitiot, A., MacAskill, M. R., & Peirce, J. W. (2020). The timing mega-study: Comparing a range of experiment generators, both lab-based and online. PeerJ, 8, e9414.
> > >
> > > [8] Cai, Yang. "How many pixels do we need to see things?." International Conference on Computational Science. Springer, Berlin, Heidelberg, 2003.
> > >
> > > [9] Ullman, Shimon, et al. "Atoms of recognition in human and computer vision." Proceedings of the National Academy of Sciences 113.10 (2016): 2744-2749.

---

> > ### Comment · Reviewer_B3mW · 2021-07-13
> > **Vision and motor responses**
> >
> > I appreciate the authors' answers. The authors seem to argue that the enormous difference in performance between previous studies and the current one could be largely attributed to the fact that observers had ample time to respond in previous studies even though stimulus duration was short (even an order of magnitude shorter than in the current study in some cases). This is possible. In this case, it is perhaps worth separating these tasks into "vision" and "motor response". Perhaps the authors may argue that this is an artificial distinction and I would be curious to know if this is the case. Then it would seem that "vision" is close to ceiling in these very simple n-way categorization tasks (all the references above). The authors are getting chance levels here at 200 or 400 ms, not because of a vision failure, but because of the difficulty in picking which one of 10 keys to press. This is of course pretty complicated, you need to translate your visual understanding into your memory of which key to press (perhaps even via image --> category i --> key j). Perhaps one of the best ways to separate "vision" from "motor" is to look directly inside the brain. Thorpe et al Nature 1996 showed categorization signals (animal or not) in about 150 ms. This is not inside the brain. Hung et al Science 2005 showed near ceiling decoding of 8 categories in about 200 ms. These are but two examples out of many. Forcing people to respond at 200 ms (or 400 ms) might perhaps mess up with the mapping to which key to press, but not with visual processing. Vision at 200 ms and even more so at 400 ms is likely to be close to 100%, in contrast to the 10% reported here.
> >
> > The 32x32 pixelation makes the problem harder and may also impact the results (how exactly from a quantitative standpoint I do not know). This is perhaps similar to the high noise conditions in Geirhos et al or the high occlusion levels in Tang et al.
> >
> > Of course, the number of categories matters a lot too. But it is hard to think that one can really get chance levels if you are at ceiling with 5 or 8.  It seems that the main problem is the motor aspect mentioned above.
> >
> > I also appreciate all the answers in part 2. All the technical details about the stimulus timing below are very clear and actually very nice!

---

> > > ### Author Response · Authors · 2021-07-14
> > > **Response to "Vision and motor responses"**
> > >
> > > ***I appreciate the authors' answers. The authors seem to argue that the enormous difference in performance between previous studies and the current one could be largely attributed to the fact that observers had ample time to respond in previous studies even though stimulus duration was short (even an order of magnitude shorter than in the current study in some cases). This is possible.***
> > >
> > > **Yes**. Thanks. For instance, for letter identification, the measured threshold contrast vs. duration is flat from 200 to 4000 ms (Pelli et al., 2006).
> > >
> > > ***In this case, it is perhaps worth separating these tasks into "vision" and "motor response". Perhaps the authors may argue that this is an artificial distinction and I would be curious to know if this is the case.***
> > >
> > > **Clarification.** We agree that in modeling it’s useful to try to treat the measured reaction time as a sum of several stages, and this has a long history in perception and decision making research. The usual model has three stages, sensory, decision making, and motor response. Some tasks allow for continuous integration of information, e.g. seeing in dim light, or more generally tasks where the target is rendered by dynamic random dots. Other tasks, which we’ll call “static”, provide all their information immediately, e.g. the static images of CIFAR-10. For integration tasks, one expects performance to improve proportionally with stimulus duration, because of accumulation of information in the sensory module. For static tasks, the sensory module requires a minimum integration time (typically about 100 ms) to overcome internal noise, and then benefits no further from longer duration. Decision making refers to the categorization of the stimulus into a response category, and this takes 100 to 500 ms, depending on the difficulty of the task. Generating the motor response (key press) takes 100 to 300 ms, depending on experience, number of possible responses, and the travel time of the finger.
> > >
> > > ***Then it would seem that "vision" is close to ceiling in these very simple n-way categorization tasks (all the references above). The authors are getting chance levels here at 200 or 400 ms, not because of a vision failure, but because of the difficulty in picking which one of 10 keys to press. This is of course pretty complicated, you need to translate your visual understanding into your memory of which key to press (perhaps even via image --> category i --> key j).***
> > >
> > > **Yes.** The task is hard because of the pixelation, added noise, and number of categories, and the very limited response time. We believe that the time pressure is strictly affecting the decision quality, not the already saturated sensory signal or the time required to make the motor response.
> > >
> > > ***Perhaps one of the best ways to separate "vision" from "motor" is to look directly inside the brain. Thorpe et al Nature 1996 showed categorization signals (animal or not) in about 150 ms. This is not inside the brain. Hung et al Science 2005 showed near ceiling decoding of 8 categories in about 200 ms. These are but two examples out of many. Forcing people to respond at 200 ms (or 400 ms) might perhaps mess up with the mapping to which key to press, but not with visual processing. Vision at 200 ms and even more so at 400 ms is likely to be close to 100%, in contrast to the 10% reported here.***
> > >
> > > **Clarification.** Yes, sort of. Can we restate it our way? In machine learning networks, sensory processing and decision making are lumped together. Furthermore when tasks are very easy it is hard to distinguish them. Our approach makes the tasks very hard by using pixelated images, adding noise, and severe time pressure on the decision. This allows us to trace the growth in performance as the decision is made.
> > >
> > > ***The 32x32 pixelation makes the problem harder and may also impact the results (how exactly from a quantitative standpoint I do not know). This is perhaps similar to the high noise conditions in Geirhos et al or the high occlusion levels in Tang et al.***
> > >
> > > **Clarification.** Yes. See above. Image resolution affects both human and machine observers. For example, Cai (2003) [8] recently studied that image content affects the resolution necessary for successful recognition in humans. Ullman et al. (2016) [9] compare neural network and human performance on MIRCs (minimal recognizable image configurations), which are smallest human-recognizable image crops. The resulting images are blurry because they represent small image regions and the authors show that networks perform much worse than humans.
> > >
> > > ***Of course, the number of categories matters a lot too. But it is hard to think that one can really get chance levels if you are at ceiling with 5 or 8. It seems that the main problem is the motor aspect mentioned above.***
> > >
> > > **Agreed.**
> > >
> > > ***I also appreciate all the answers in part 2. All the technical details about the stimulus timing below are very clear and actually very nice!***
> > >
> > > **Thank You!**

---

### Official Review · Reviewer_iMX8 · 2021-07-05
**More suitable for a cogsci venue**

**Rating:** 5
**Confidence:** 4

**Strengths:**

There are several strengths of this paper.

S1. It is motivated by an important problem.  Using neural networks as computational models of perception or cognition is useful with potential benefits to our understanding and practice of neural networks in engineering and sciences.

S2. The paper provides a relatively thorough evaluation of the models considered for the human-like speed-accuracy performance trade-off.

S3. It is written well, overall, and easy to follow.


**Weaknesses:**

There are also a few weaknesses of the paper.

W1.  The contributions to earlier work, particularly over [6], are not substantial. Also, it is unclear how statistically significant the differences in correlations across models found in the paper are.

W2.   Given W1, one would expect the paper to focus on data collection and analysis. However, the collected data is small scale and the paper focuses on evaluating different dynamic neural networks, instead of data curation and analysis. For example, one hour is at the long end of crowdsourced elicitation sessions and presents potential issues on the benchmark data quality.

W3. The baseline model ConvRNN was originally trained on the ImageNet dataset and,  given that images are upsampled here, it is unclear why the  ImageNet dataset was not used. CIFAR-10 might be too easy for the models considered to show a performance range.


**Additional Feedback:**

I find the paper adding to our understanding of how neural networks can be used as computational models to study the accuracy-speed trade-off in human object detection. However, its contributions, both in modeling and data collection, are limited.  It'd be a better fit to be published at a cogsci venue as an update to [6].

**Post Rebuttal**  I appreciate the authors’ detailed response and the revision. I also enjoyed the style of their response. Nonetheless, given its contributions, I still think this work will be better assessed and appreciated at a cogsci venue.


**Clarity:**

The paper is written well and straightforward to follow.

Minor.
There seems to be a mismatch between Table 1 (20+7+8=35) and Sec 3.2 (33) on the number of MTurk participants.

**Correctness:**

The paper misses an analysis of data quality, which can be an issue given that each MTurk session took about an hour to finish. At a minimum, the variance among participants across task conditions should be reported.
Another issue is that the paper doesn't also report the statistical significance of correlations reported as well as the statistical significance of the differences between the models compared using these correlations.  It is unclear, for example, whether the difference between the ConvCNN and MSDNet models in predicting human behavior is statistically significant or not.


**Documentation:**

The dataset documentation has almost all the pieces for accessibility and reproducibility. One thing I couldn't find is how to reproduce the MTurk object recognition experiment.
Also, the links for the deployed online experiments  give the following error:
'A problem occurred: It's not allowed to run this study (ID: 84) in this batch (ID: 85) with a worker of type "General Multiple".'

**Ethics:**

The paper doesn't seem to introduce additional ethical concerns beyond the risks shared across standard machine learning benchmarks and applications.

**Relation To Prior Work:**

The paper can give an improved discussion on how the current work differs from [6], which introduces the baseline ConvRNN model and most of the methodology used here, and, in this context, why the ImageNet dataset was not used for the current study.

**Summary And Contributions:**

The paper evaluates three adaptive dynamic neural networks for their correlation with the speed-accuracy tradeoff of human object recognition in 2D images.  For this, it first collects a benchmark dataset elicited from crowdworkers under various transformation conditions using the CIFAR-10 dataset.  Then it compares the ability of the rCNN (ConvRNN), MSDNet, and SCAN models to capture the speed-accuracy trade-off exhibited in the crowdsourced perceptual data and shows the MSDNet model's predictions exhibit the most human-like speed-accuracy tradeoff.

---

> ### Author Response · Authors · 2021-07-12
> **Response Part 1**
>
> ***It is motivated by an important problem. Using neural networks as computational models of perception or cognition is useful with potential benefits to our understanding and practice of neural networks in engineering and sciences.***
>
> Thank you.
>
> ***The paper provides a relatively thorough evaluation of the models considered for the human-like speed-accuracy performance trade-off. It is written well, overall, and easy to follow.***
>
> Thank you.
>
> ***W1. The contributions to earlier work, particularly over [6], are not substantial.***
>
> *Done.* Thank you. Unlike Spoerer et al. [6], we show that anytime-prediction networks (MSDNet, SCAN) are promising new models for the human speed-accuracy tradeoff and exhibit similar human-like behavior as rCNN (from Spoerer et al. [6]) which was developed specifically as a model for human reaction time.
> In doing so, we introduce:
>
> *  *New Metrics*: Our work focuses on comparing human and model performance as a function of reaction time in contrast to Spoerer et al [6] which compares only reaction time for prediction. We apply two metrics: a) correlation of performance as function of reaction time, with the average human; b) performance range. Unlike those used in [6], these metrics capture both performance and reaction time and hence allow for a more robust evaluation of the speed-accuracy tradeoff exhibited by humans and models. Lines 268-271 (Section 5.2) and lines 305-308 (Section 6) in the updated manuscript now better spell out this aspect of our contribution.
> *  *A New Dataset*: We contribute an open-access dataset of human speed-accuracy trade off in object recognition that is suitable for comparison with machine performance. Performance spans a wide range (20% to 90%), which is way beyond the range of most neural net recognition models. We adjust task difficulty using several popular manipulations: color vs. grayscale, adding noise, and blurring.
>
>
>
>
> __*Also, it is unclear how statistically significant the differences in correlations across models found in the paper are.*__
>
> *Done.* We have now performed statistical analysis of our results. We ran paired, 2-tailed t-tests using p = 0.05 (with Bonferroni correction, similar to [6]) to evaluate whether the difference in correlation between different models is significant and text (Lines 296-299) in the paper has been updated accordingly. Our main conclusion from the correlation plots still holds: MSDNet, an anytime prediction network shows human-like behavior similar to rCNN (ConvRNN) from Spoerer et al. [6] which was developed specifically as a model for the human speed-accuracy tradeoff.
>
>
> __*W2: Given W1, one would expect the paper to focus on data collection and analysis. However, the collected data is small scale and the paper focuses on evaluating different dynamic neural networks, instead of data curation and analysis. For example, one hour is at the long end of crowdsourced elicitation sessions and presents potential issues on the benchmark data quality.*__
>
> *Done.* We now ran a Power sample size test (see Hulley et al. 2013) for power = 80% and p=0.05 and determined that the size of our dataset is large enough to draw all conclusions reported in the paper. The paper has been updated with this statement on lines 296-299
>
> *Clarification.* Our goal is to create an open-source benchmark and present anytime prediction networks as a promising alternative demonstrating more human-like behavior than previous work (rCNN, from Spoerer et al. [6]). Our survey design was based on the previous work by McElree & Carrasco [5], where 4 observers participated in a total of 20 approximately 75 min sessions (now mentioned in Section 3.2, lines 124-125). Furthermore, our lab tests participants for thousands of hours each year, typically with one hour sessions and we have many times compared data at the beginning and end of a session and never noticed a significant difference.
>
> __*W3. The baseline model ConvRNN was originally trained on the ImageNet dataset and, given that images are upsampled here, it is unclear why the ImageNet dataset was not used.*__
>
> *Explanation.* First, we used CIFAR-10 instead of ImageNet because the images are smaller allowing much faster training of networks. Second, all of these dataset have too many categories to test on one single human. One has to choose. Spoerer et al. used the large ImageNet set, but reduced it to two categories (animate vs. inanimate), which makes the task for like detection (of life) than object recognition. Humans doing detection tasks quickly learn to focus on just the defining difference, unless object recognition which typically draws from the whole object. Instead, we used all of CIFAR-10, with its ten categories. We believe this better captures human object recognition performance. This choice is now spelled out in Section 3.1 lines 104-112.

---

> > ### Author Response · Authors · 2021-07-12
> > **Response Part 2**
> >
> > __*CIFAR-10 might be too easy for the models considered to show a performance range.*__
> >
> > *Explanation.* Yes, CIFAR-10 is easy and risks getting a ceiling effect. However, we restricted the network resources to attain less than perfect performance, and added noise or blur to bring it down further, allowing us to measure the speed-accuracy tradeoff. We now explain this in lines 263-265
> >
> > __*The paper misses an analysis of data quality, which can be an issue given that each MTurk session took about an hour to finish. At a minimum, the variance among participants across task conditions should be reported.*__
> >
> > *Done.* Plots showing variance in reaction time and accuracy across participants will be added to the Appendix. This will be in addition to Figure 1 in the existing version of the paper which shows variation of accuracy across participants for each timestep, in the form of separate trade-off curves.
> >
> > __*There seems to be a mismatch between Table 1 (20+7+8=35) and Sec 3.2 (33) on the number of MTurk participants.*__
> >
> > *Done.* Oops. There were 35 participants in total, as was already reported in Table 1. This has now been resolved.
> >
> > __*The dataset documentation has almost all the pieces for accessibility and reproducibility. One thing I couldn't find is how to reproduce the MTurk object recognition experiment. Also, the links for the deployed online experiments give the following error: 'A problem occurred: It's not allowed to run this study (ID: 84) in this batch (ID: 85) with a worker of type "General Multiple".'*__
> >
> > *Done.* Thank you for reporting this error. We have made the necessary modifications to the dataset documentation.

---

### Author Response · Authors · 2021-07-12
**Thank You!**

**Dear Reviewers**:
We thank all three reviewers for their constructive comments and suggestions. As requested, we have performed statistical analyses of the results, and polished Methods to be more clear. Details below.

We are sharing with you current drafts of our response letter and the revised manuscript. It includes many references to changes in the manuscript that are still being implemented. We hope that this complete response letter will help you assess whether we’ve missed any of your points.

**Ajay, Omkar, Elena, Najib, Denis**

---

### Decision · Program_Chairs · 2021-07-26

**Decision:**

Reject

**Comment:**

The paper presents a very interesting problem, a moderate size dataset, and an associated comprehensive evaluation. Reviewers share concerns regarding novelty, and in particular regarding the size/impact of the dataset. It is appreciated the effort authors did by providing all justifications and paper updates as requested by the reviewers. However, still there are concerns regarding the size/impact of the dataset..